# The Volumetric Wear Assessment of a Mining Conical Pick Using the Photogrammetric Approach

**DOI:** 10.3390/ma15165783

**Published:** 2022-08-22

**Authors:** Jan Pawlik, Aleksandra Wróblewska-Pawlik, Michał Bembenek

**Affiliations:** Faculty of Mechanical Engineering and Robotics, AGH University of Science and Technology, A. Mickiewicza 30, 30-059 Kraków, Poland

**Keywords:** conical pick, photogrammetry, mining, hardfacing, photogrammetric method

## Abstract

The rapid wear of conical picks used in rock cutting heads in the mining industry has a significant economic impact in cost effectiveness for a given mineral extraction business. Any mining facility could benefit from decreasing the cost along with a substantial durability increase of a conical pick; thus, the hardfacing method of production and regeneration should be taken into account. In order to automatize the regeneration, the wear rate assessment is necessary. This paper presents a methodology used to create a 3D photogrammetric model of most of the commercially available tangential-rotary cutters in their before and after abrasive exploitation state. An experiment of three factors on two levels is carried out to indicate the proper setup of the scanning rig to obtain plausible results. Those factors are: light level, presence of polarizing filter and the distance from the scanned object. The 3D scan of the worn out specimen is compared to the master model via algorithm developed by the authors. This approach provides more detailed information about the wear mechanism and can help either in roadheader cutting head diagnostics or to develop a strategy and optimize the toolpath for the numerically controlled hardfacing machine.

## 1. Introduction

Wear of any kind of industrial tool is an inevitable phenomenon. The tools are subjected to many divergent wear mechanisms [1], depending mainly on the material selection and working environment [2]. The mining industry can be considered to be one of the extreme cases, considering the high loads and abrasive conditions to which the “end tools” are subjected. Considering tool wear measurement, one can stumble upon many approaches. Some of them are based on measurement of the response for a vibratory excitation [3,4]; others utilize mass measurement or harness a vision system [5].

A conical pick (sometimes named “tangential-rotary cutter”) is a good example of a mining tool, designed to withstand harsh working conditions and crush hard rocks, yet it still tends to lose its primal shape and durability after a certain operational period. Replacement of the tool is problematic and expensive, but is sometimes needed even after a few hours of work, which leads to the need for remanufacturing of the tool [6,7]. The main challenge of the regeneration process of worn out tools at the first stage is the difficulty of classifying its wear rate and deciding whether the tool can still be in use or should be discarded. Whenever the cavities are excessive, it is crucial to determine the possibility of regeneration. If the particular pick is suitable for reconstruction, the next step is to develop a strategy for fabrication of a hardfaced coating [8]. This process is usually performed by skilled welders manually; however, today one can incorporate a numerically controlled hardfacing machine.

Previous research such as [9,10,11] proposes a method to define the C2 parameter which determines the wear rate that relies on the conical pick’s loss in weight after the machine tool process. The method consists of disassembly of picks and measurement of their mass and the volume of output cut material obtained during the work of picks subjected to testing.

Using only a weighing sensor and measurements of the cut mineral makes it a suitable method of obtaining the parameter for rough analysis in difficult environment conditions such as mining facilities. This, however, does not bring about any further information on spatial deformation but only on the overall wear when compared to the master model, which can be misleading, especially during analysis of tools that have local cavities correlated to tools that have uniform abrasions (Figure 1). The products of plastic deformations of the working part can still be attached to the pick’s body; therefore, theoretically, they can also add distortions to the mass measurement. Additionally, the tungsten carbide tip can have up to double the density of the steel, which can make mass measurements even less informative [12].

In [13], the quality assessment procedure for picks was presented. Apart from calculating the C2 parameter, classification takes into account other criteria. Some of the measurements described in the paper require laboratory conditions; therefore, this process cannot be conducted in a difficult environment.

The next method of determining wear level is to analyze the mining power consumption signal [14]. The classification is carried out with the use of a wavelet transform for noise reduction of measurement data of the mining power signal and artificial intelligence. Despite the results being satisfactory, attention should be paid to selection of the base wavelet, which influences the obtained final results. This means that every sample should be analyzed by a person who should decide on the mathematical operator of the calculations, which can complicate the process.

Another method of obtaining the digital parameters of tool wear is image processing [15,16]. This gives some or all of the information that can be gathered by the human eye, creating a digital model that can be processed further on. For an industrial company that has the possibility to conduct LIDAR scanning, such a model can be obtained using the methodology described in [17,18,19]. However, this type of measurement uses the method of physically hitting a model’s feature with light and measuring the reflection so that the texture of the object is not included in the results.

The solution to this problem that was chosen by the authors is the usage of a photogrammetry process that relies on images captured by a camera to reconstruct the 3D model coordinates using image overlap [20]. The photogrammetric model method brings forth enough information for conical picks regeneration, as the output data include information about local cavities of the operating part, its texture and its ability to be automated. This method does not require any expensive equipment or personnel training.

There are many photogrammetric methods, but one can distinguish three main ones considering the dynamic of an object that is projected and the cameras that are needed for image acquisition [21]:Static object, many cameras are triggered at the same moment, placed around the object;Static object, one camera moving around the object while taking pictures;Rotating object, static camera.

The authors have chosen the last method because of feasibility of the automation and low cost of the module capture setup. Data acquisition for photogrammetry should be conducted in a controlled environment [22,23,24] to ensure that the image quality is not influenced by factors such as extensive light, transparent surfaces, light reflections and surrounding objects. To do so, the choice of light source should be adapted to the object surface as well. It is best when the object diffuses light which is common as the outcome of the object’s roughness. For metallic surfaces that can reflect light easily, the source light should be uniform and diffused [25]; additionally, cross-polarization and chromatic filters may enhance the results [26,27,28]. The baseline/distance ratio, i.e., the distance between two camera positions to the distance between the camera to the object ratio should be between 1/15 and 1/20 [29]. Crucial parameters that have a great impact on the suitability for photogrammetric synthesis are:Exposure time (shutter speed);Aperture;Depth of field;Sensitivity of light to camera (ISO).

Those parameters should be selected as a compromise, to result in capturing sharp images that are the appropriate input for the processing algorithm. The use of a tripod usually facilitates the process, especially for underexposed objects or long exposure time camera settings. The aperture parameter and shutter speed should be set according to ISO, the value of which should be the lowest possible to avoid digital noise in the images.

In the current study, the authors attempt to build a low-cost photogrammetric scanning setup for conical picks and estimate the process capabilities and general robustness, where the main goal of this paper is to develop and study a potential pick wear assessment algorithm scheme. The acquired data can serve many purposes, both as a practical tool for mining consumable evaluation and regeneration and as a mathematical model verification benchmark. The results of the scanning can be compared with the numerical models, such as EDEM approach by Liu et al. [30] or the peak cutting force model built by Kuidong et al. [31] or other tangential-rotary cutter theoretical approaches [32,33,34,35,36,37,38,39]. All of the cited research could benefit from introducing the method of confirmation of calculated mechanical properties with the output geometry of an exploited tool, existing in reality.

In the following sections, the authors describe the studied conical picks at various stages of wear and the scanning setup, both from the hardware and software point of view. Afterwards, the aligning and scaling algorithm is presented and tested on the divergent geometry. Lastly, the authors test robustness of their method with the help of the Taguchi Orthogonal Arrays design, in which they estimate which factor (light level, presence of polarizing film and the distance from camera to scanned object) has the biggest impact on the process stability.

## 2. Materials and Methods

### 2.1. Studied Specimen

In this study, the authors examine road cutting conical picks, manufactured from 34CrMo4 steel with geometry that is compliant with Figure 2. The tool in Figure 3 noted as CP0 is a brand-new pick, which will serve as a reference specimen for the degraded tools which had contact with abrasive rock.

### 2.2. Measurement Setup

The setup consisted of the phone camera, motorized turntable and the lightbox (Figure 4) and a computer with the required software installed.

In order to ensure the stable movement with constant velocity, the authors designed and manufactured a special rotary mount. This turntable was built from a stator base, a bearing and a belt-driven rotor pick holder. The motion was created by a bipolar stepper motor, controlled by a TMC2208 stepper driver (Trinamic GmBH, Hamburg, Germany), connected to an Arduino Nano microcontroller (Arduino, Sommerville, MA, USA). The pick holder was rotating at *n* = 0.6 rot/min. The golden and silver symbols on each of the 12 walls of the rotor holder serve as the reference pattern for the 3D reconstruction algorithm—if the reader was to rerun the experiment, he can apply any kind of non-repeating pattern. That pattern brings forth an especially valuable contribution when one scans a reflective and undeformed revolving solid.

### 2.3. Data Acquisition

Instead of image acquisitions, the authors decided to capture a video and split it into 60 frames. The video clip was captured with a 12.2 MP Sony IMX333 sensor (Sony, Tokyo, Japan), integrated in a Samsung Galaxy S8 smartphone (Samsung, Suwon, South Korea). The authors did not use the Samsung proprietary video-capturing software due to the possible presence of unknown filters and video enhancements; instead, they utilized IVCam software (e2esoft, Shanghai, China) and the capture was executed via a client installed on the computer. Measurement series were conducted using groups of parameters of values from the settings limit in Table 1. Selected parameters groups are described in Section 2.4.

### 2.4. Image Processing

The authors used open-source software Meshroom, version 2021.1.0 (AliceVision Association, Paris, France) to obtain 3D models. It is a program based on the AliceVision framework with a specific pipeline for a project. Single steps of the pipeline consist of 3D model calculating algorithm steps:Natural feature extraction [40,41,42,43,44];Image matching [45];Features matching [46];Structure from motion [47];Depth maps estimation [48];Meshing [49];Texturing [50].

The first step was to extract 60 frames from captured videos. To do so, authors prepared the MATLAB (MathWorks, Natick, MA, USA) script that enabled saving images in a certain directory. Afterwards, a new project in Meshroom was created and the images saved were determined as the input for the algorithm.

The results of consecutive steps are presented in Section 3. The final step was to save the 3D model of the working part of the scanned picks in .stl format and analyze its geometry in the MATLAB environment (Section 2.5).

### 2.5. Statistical Analysis

The optimization process was planned as presented in Table 2. The first step was to choose appropriate parameter values for parametric analysis. The ISO value of 200 was set a priori and the light levels were set with the use of LED lights switch. The rest of the parameters were selected accordingly, for two of the light levels as in Table 3. The light level was measured at the position of the conical pick’s tip with the GH59-14759A light sensor.

The impact of light level, distance to object and polarizing filter was analyzed by performing eight tests on the master model to choose optimal settings for the process. Optimization consisted of analysis of the influence of three factors with two levels (L8) on the resulting output function values that determined the model quality. 

The process was optimized for best output model accuracy. For this purpose:

The algorithm for cavities classification was made and objective Function (1) was constructed for maximization target.
(1)$$f\left(n_{i},n_{p},a\right)=\frac{n_{i}}{N}+\frac{n_{p}}{P_{max}}+a_{m},$$
where:*n_i_*—number of images classified as proper, *N* ≥ *n_i_* ≥ 0;*N*—number of all input images, *N* = 60;*n_p_*—number of characteristic points matched, *P_max_* ≥ *n_i_* ≥ 0;*P_max_*—number of maximal amount of characteristic features matched points achieved;*a_m_*—accuracy of the 3D model, *a_m_* ⊂ {0; 0.5; 1}.

The *a_m_* parameter was determined by the authors following the rules: if the 3D model properly projects the geometry of the conical pick and is the suitable input for the wear classification algorithm, the *a_m_* value is equal to 1. If it is conditionally suitable, the *a_m_* value is equal to 0.5. If the model is improper, the *a_m_* value is equal to 0.

### 2.6. Wear Classification

Five samples of conical picks after exploitation were evaluated. Their 3D models were examined in the geometry analysis process. Their symmetrical wear was stated (2) as below.
(2)$$S=\{\begin{array}{c} 1;\sigma_{A}\leq 0.3A_{M}\\ 0;\sigma_{A}>0.3A_{M} \end{array},$$
where:*S*—symmetry determinant;*σ_A_*—standard deviation of cross-sections of 3D model;*A_M_*—area of the cross-section of the master model.

## 3. Results

### 3.1. Parametric Optimization

The parametric optimization process for obtaining the model of the conical pick resulted in the eight 3D models presented in Table 4. In Table 5, the properly and improperly projected picks are presented. Observations of output are described below:Run 3, 7—geometry projected on a plane, cameras detected improperly. In both runs, image is far from camera.Run 1, 5—geometry is generally proper, the carbide part has geometry artifacts from the reflected line as it is smooth material.Run 4, 8—no valid initial pair found automatically.Run 2, 6—geometry is proper.

Calculation of signal to noise ratio according to the rule “the-larger-is-better”:


(3)
$${\mathrm{SN}}_{i}=-10*\log_{10}\frac{\sum_{1}^{n}\left(\frac{1}{Y^{2}}\right)}{n}$$


Calculation of impact of each factor on the subjective function value:


$$F(A)=\frac{f5+f6+f7+f8}{4}-\frac{f5+f6+f7+f8}{4}=0.0807$$



$$F(B)=\frac{f3+f4+f7+f8}{4}-\frac{f5+f6+f1+f2}{4}=-2.0303$$



$$F(C)=\frac{f2+f4+f6+f8}{4}-\frac{f1+f3+f5+f7}{4}=-0.4458$$


Calculation of the resistance of each factor to noise:


$$\mathrm{SN}(A)=\frac{\mathrm{SN}5+\mathrm{SN}6+\mathrm{SN}7+\mathrm{SN}8}{4}-\frac{\mathrm{SN}5+\mathrm{SN}6+\mathrm{SN}7+\mathrm{SN}8}{4}=0.2439$$



$$\mathrm{SN}(B)=\frac{\mathrm{SN}3+\mathrm{SN}4+\mathrm{SN}7+\mathrm{SN}8}{4}-\frac{\mathrm{SN}5+\mathrm{SN}6+\mathrm{SN}1+\mathrm{SN}2}{4}=-57.7305$$



$$\mathrm{SN}(C)=\frac{\mathrm{SN}2+\mathrm{SN}4+\mathrm{SN}6+\mathrm{SN}8}{4}-\frac{\mathrm{SN}1+\mathrm{SN}3+\mathrm{SN}5+\mathrm{SN}7}{4}=-50.1441$$


Parametric optimization results are shown in Figure 5 and the parameters’ influence on the objective function value is shown in Figure 6.

### 3.2. Features Extraction

Individual model features were determined using MATLAB scripts to perform calculations on the .stl models. The first step was to center the data (3D points cloud) at zero. Next, the direction of most variance and rotation of the data was found to align it to the Z axis and translate it afterwards so that all of the data points are aligned so that z values are greater than 0 (Figure 7).

As can be seen in Figure 7, the resulting geometry may have a different orientation along the X axis. To change the pick’s position so that the carbide has the X coordinate equal to 0 and the rest of its geometry lies on the right side of the axis, linear regression of density in the domain of the X coordinate was calculated and the slope value was checked. If a model is represented by regression with a slope value bigger than zero, all of the points were rotated along the Y axis (Figure 8).

The next step was to find the X axis coordinate of the pick’s holder end and then to scale the model. It was made using calculation of data point density and the scale factor comparing maximal diameter values of the holder and its projected geometry (Figure 9).

The symmetry was calculated for 37 areas of cross-sections of the model with 10° of difference between: ϕ = [0°, 10°, …, 350°, 360°]. The 2D boundary of the model was determined and its area was calculated. Additionally, the plastic deformation area was defined, determining each sum of cross-section boundaries of the master and the rest of the picks (Figure 10). Afterwards, the difference between the summed boundary and the pick’s original boundary was calculated (Table 6). Figure 11 presents the aggregated results for the examined picks.

Cross-section boundary points of the pick nr 4 are considered to be worn out and are not suitable for regeneration.

As expected, the CP0, being the reference specimen, has low deviation and the highest average cross-section value. The eccentrically placed mean value (red line) in the CP2, CP3 and CP4 provides information about the asymmetrical wear of the picks. The bigger the box, the bigger the value of asymmetry.

## 4. Discussion

The scanned conical picks are typical examples of a set of tools removed during maintenance of the cutting head. If they were to be rated for their further usability, one could divide them among subsequent categories:Eligible for use: CP0, CP1;Eligible for hardfacing: CP0, CP1, CP2;Catastrophic wear: CP3, CP4.

The photogrammetric measurement of volume followed by the symmetry assessment script provides plausible results, especially considering its low cost and high reliability. The authors believe that implementing this method in the mining repair plants could be beneficial indeed. This approach could also serve as a basic quality control unit in a mining tool factory or another facility which manufactures parts with an axis of symmetry. Yet, it is noteworthy to consider the surface roughness (or reflectiveness, to be more precise) of the scanned part. The very reflective, polished objects with the R_a_ parameter below 2.5–5 µm cause some issues, namely, unexpected bumps or cavities in the place of a very bright spot. The solution for those issues is either to have those surfaces dulled with talc or another powder or to incorporate a different method for obtaining the 3D geometry.

Overall, this approach to tool wear characterization is relatively easy to use and provides much information about the wear mechanism. The current algorithm used in this paper will be applicable only to objects which are solids of revolution, since one of the steps is to find the axis of symmetry and align scanned objects according to the found axis. Nevertheless, after some modifications, a similar approach can be utilized to assess the wear rate (or even metrological compliance of the physical object with the designed virtual model) of other tools or parts. The setup that the authors used was supposed to be affordable for most populations, thus some efficiency-related concerns were a trade-off. Yet, increasing the efficiency is theoretically simple, since one has only to add more cameras to capture the images “at once”, with little to no scanned object rotation.

One of the strengths of the current setup is that it separates the scanned object from the environment, being a method of scanning which is—quite literally—as robust as the walls of the lightbox. On the other hand, the necessity of the lightbox limits the maximum size of the scanned part.

## 5. Conclusions

Described methods of classification of the picks’ wear are presented in the table below (Table 7).

By analyzing the presented methods from previous research, it can be stated that the parametric factors method is not suitable to be implemented in harsh conditions, i.e., in mining factories. The parametric factors method consists of laboratory measurements, including microscopic image analysis, which cannot be conducted in such conditions. Parametric factors, LIDAR and fuzzy neural network methods are not feasible to automate because of the large number of various processes included when it comes to the parametric factors method and the need for selection of the mathematical operator for a sample when it comes to the neural network method. The LIDAR method does not bring forth information about texture that can be the input for the algorithm of classification. The crucial disadvantage of this method is the poor availability of equipment compared to photogrammetry. The chosen method tends to be universal, as it can be performed using a phone camera. The C2 parameter assessment is still the quickest approach to conical pick wear rate evaluation; nevertheless, it does provide the user only with basic information about the wear characteristics.

The most influential of the studied parameters appeared to be the distance between the camera and the object. Nevertheless, the distances in this study were adjusted to the quality and focal point of the particular lens. Having an image-capturing device with a lens able to zoom in without any image distortion, the distances could vary significantly. The presence of the polarizing filter appeared to decrease the number of bright reflections, which resulted in 3D reconstructions with greater fidelity.

The goal of further work is to reduce the scanning and data processing time. Additionally, the authors plan to build a portable scanning device, which will enable the mining company maintenance team to gather the data on-site. This task, however, would require some extra steps in order to meet the requirements of underground heavy industry, e.g., a dustproof casing, spark-proof design of the drive and electronics and perhaps a conical pick initial cleaning device.

Another goal for future study might be connected with the advantage of photogrammetry over LIDAR or other laser-based scanning techniques, namely, the texture analysis. Since photogrammetry provides some otherwise lost information on the color of the surface of the scanned object; the textured 3D file could be subjected to more sophisticated analysis, such as hardfaced material overheat detection. Another great use of the method studied by the authors is the possibility to scan and instantly send a 3D textured file to a locally unavailable wear expert for analysis. 

The algorithm, after development, could also serve as a low-cost linear and angular measurement system for the manufactured tools. In the case of conical picks, the geometry of the working part makes it difficult to utilize conventional means of measurement.

## Figures and Tables

**Figure 1 materials-15-05783-f001:**
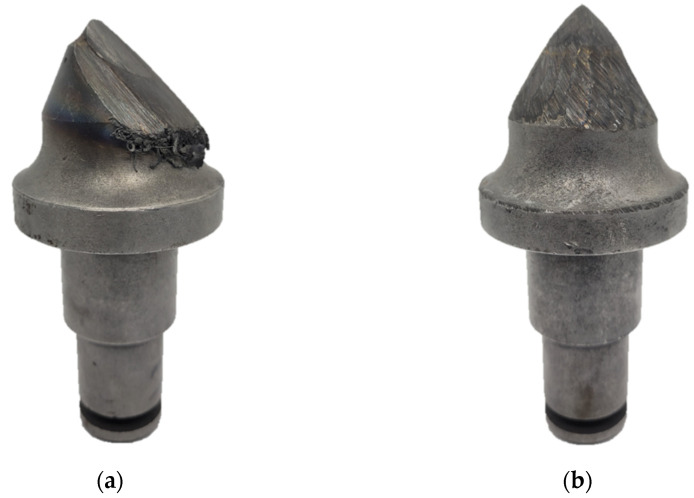
Tools with local cavities—(**a**); uniform abrasions—(**b**).

**Figure 2 materials-15-05783-f002:**
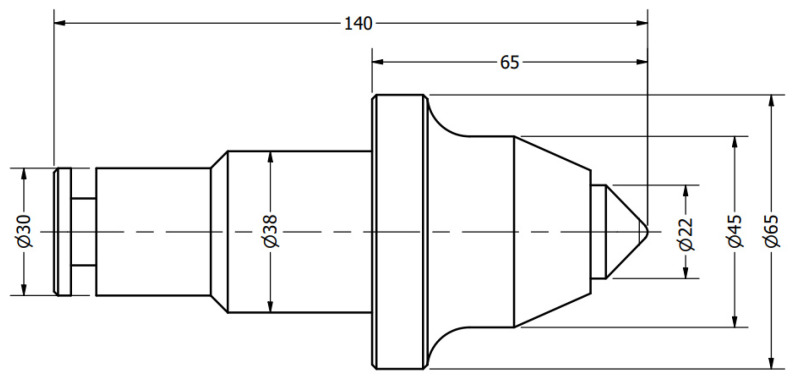
Technical drawing of the inspected conical pick. This geometry corresponds with the master model (CP0).

**Figure 3 materials-15-05783-f003:**
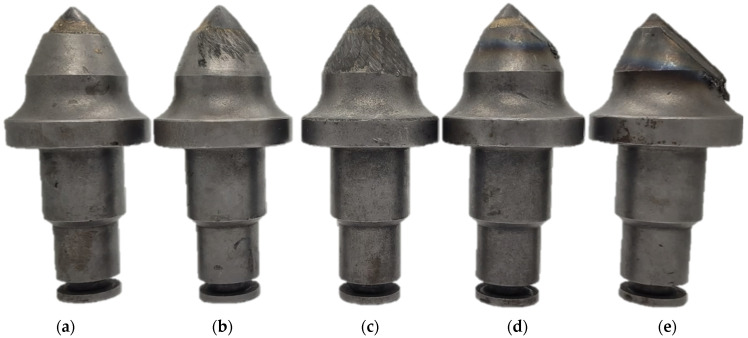
Image of all inspected conical pick specimens. The authors annotated them in the following manner: (**a**)—CP0 (master model), (**b**)—CP1, (**c**)—CP2, (**d**)—CP3, (**e**)—CP4.

**Figure 4 materials-15-05783-f004:**
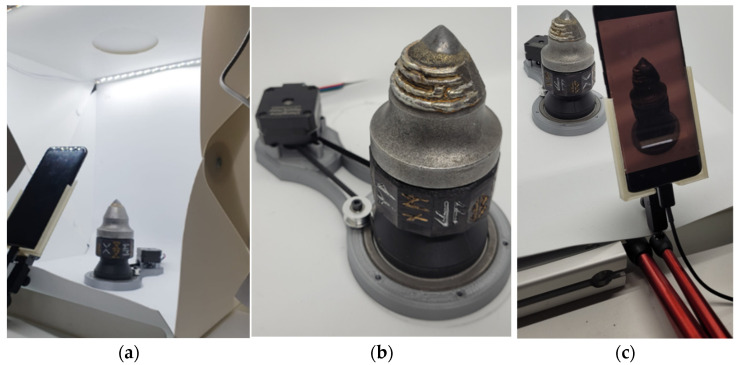
Measurement setup: (**a**)—lightbox, (**b**)—turntable, (**c**)—phone stand with 3D printed holder. Please note that the above images present a pre-hardfaced pick, which is not included in this study.

**Figure 5 materials-15-05783-f005:**
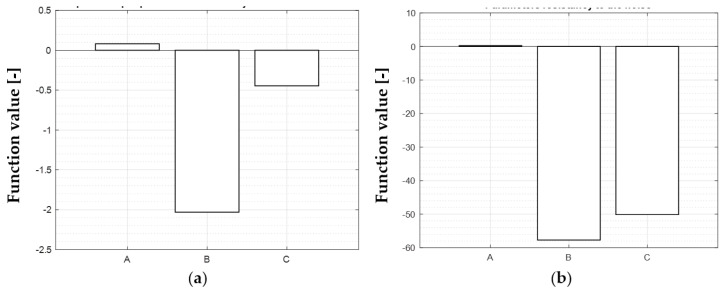
Parametric optimization results; (**a**)—impact of input parameters with impact increasing: B, C, A; (**b**)—input parameters’ resistance to noise with resistance increasing: A, C, B.

**Figure 6 materials-15-05783-f006:**
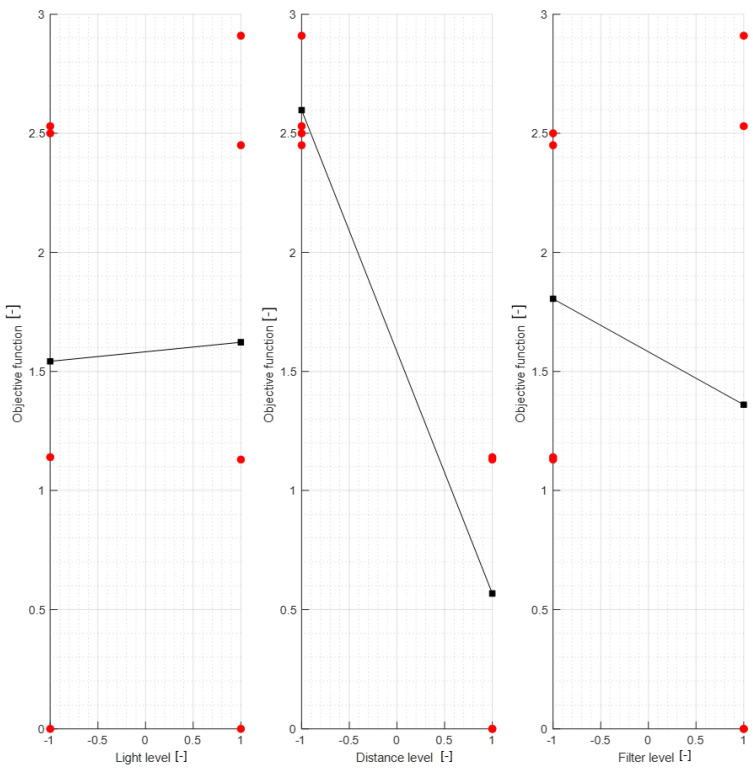
Parameters’ influence on the objective function value. The red points correspond to the values on the boundary conditions before averaging.

**Figure 7 materials-15-05783-f007:**
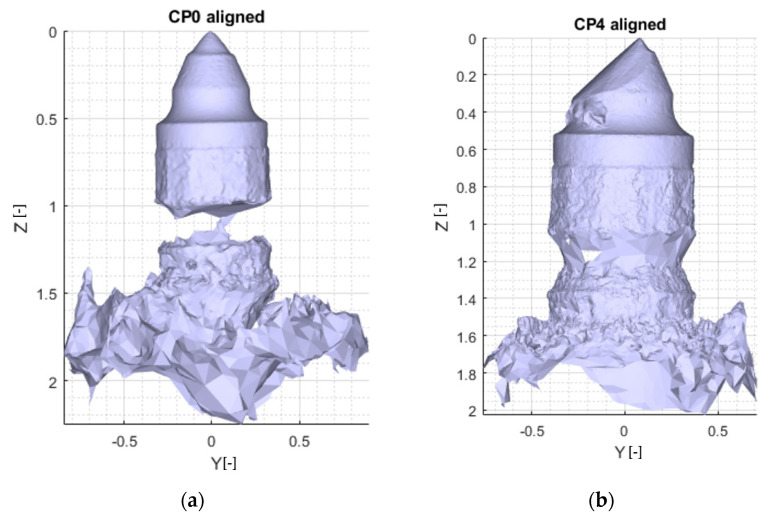
Aligned .stl model of the pick; (**a**)—master model, (**b**)—pick after exploitation.

**Figure 8 materials-15-05783-f008:**
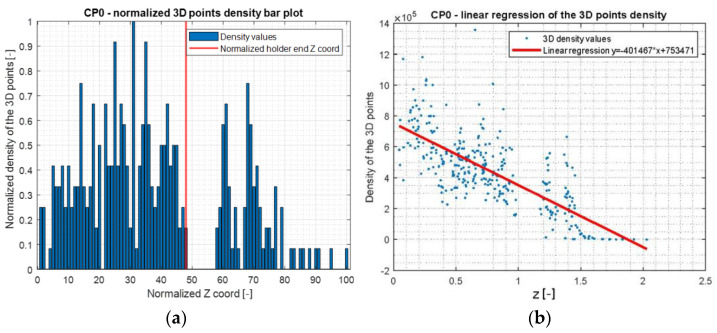
Three-dimensional space density; (**a**)—bar plot in Z coordinate domain, (**b**)—regression.

**Figure 9 materials-15-05783-f009:**
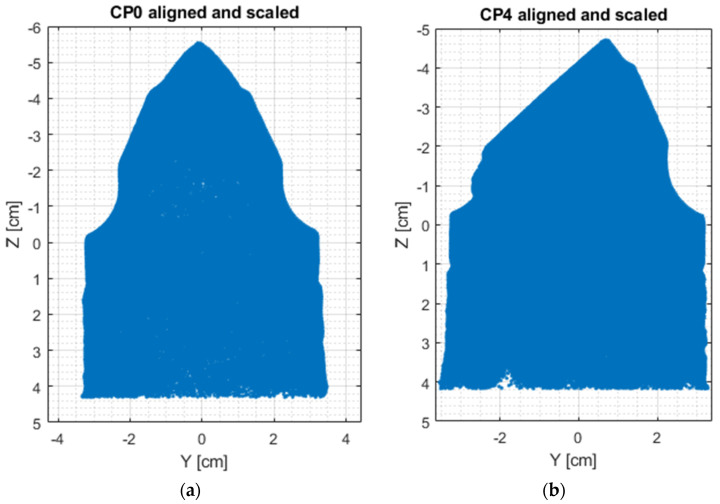
Scaled and cropped data points; (**a**)—master model, (**b**)—pick after exploitation.

**Figure 10 materials-15-05783-f010:**
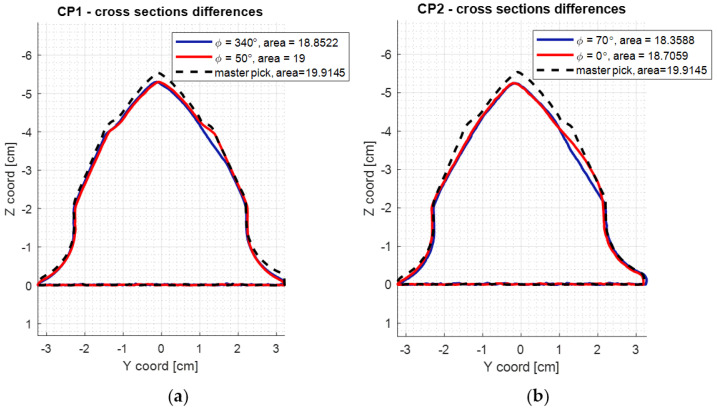
The difference between the minimal and maximal cross-section area of the scanned picks. (**a**) shows the min/max cross-sectional area of CP01 pick, (**b**) of CP02, (**c**) of CP03 and (**d**) of CP04. Their physical representation is shown in Figure 3.

**Figure 11 materials-15-05783-f011:**
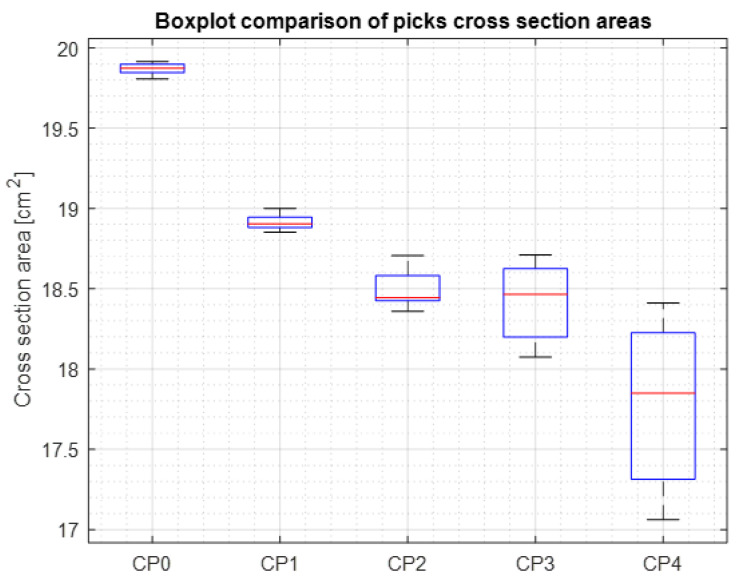
Cross-section areas of picks.

**Table 1 materials-15-05783-t001:** Camera and setup limits.

Light Level [lux]	Shutter [s]	ISO	Distance from Camera to Object [mm]	Polarizing Filters	Aperture
102–288	1/1000–1/20	200	155–250	yes/no	f/1.7

**Table 2 materials-15-05783-t002:** Experimental scenarios.

Run	Light Level	Distance from Camera to Object	Polarizing Filters
1	−	−	−
2	−	−	+
3	−	+	−
4	−	+	+
5	+	−	−
6	+	−	+
7	+	+	−
8	+	+	+

**Table 3 materials-15-05783-t003:** Camera and setup settings.

	Light Level [lux]	Distance from Camera to Object [mm]	Polarizing Filters Included
Lower limit (−)	102	155	No
Upper limit (+)	288	250	Yes

**Table 4 materials-15-05783-t004:** Parametric optimization results.

Run	Light Level (A)	Distance (B)	Polarizing Filter (C)	Randomized Trial [-]	Images Classified *n_i_*	Points Matched *n_p_*	Accuracy *a_m_*	Objective Function *f_i_*	SN_i_
1	−	−	−	1	60	5806	0.5	2.5	7.9588
2	−	−	+	6	60	3050	1	2.525	8.0452
3	−	+	−	4	59	886	0	1.136	1.1076
4	−	+	+	8	0	0	0	0	−100
5	+	−	−	2	60	5529	0.5	2.452	7.7904
6	+	−	+	5	60	5262	1	2.906	9.2659
7	+	+	−	3	60	734	0	1.126	1.0308
8	+	+	+	7	0	0	0	0	−100

**Table 5 materials-15-05783-t005:** Properly projected geometry, f = 2.525; improperly projected geometry, f = 1.126.

Run	Mesh	Texture	Structure from Motion
2	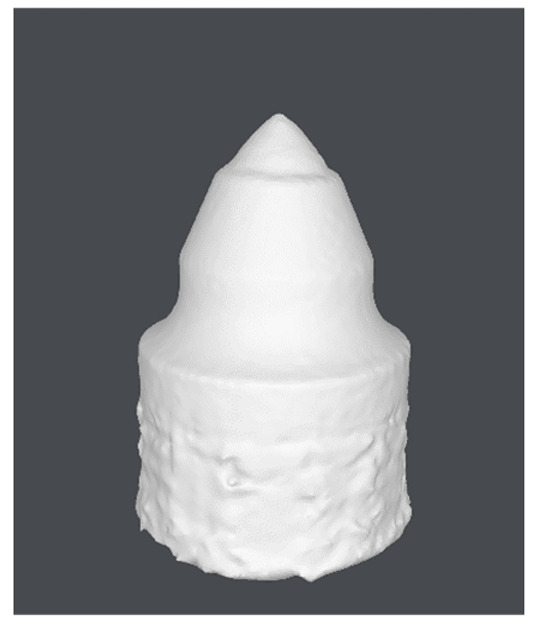	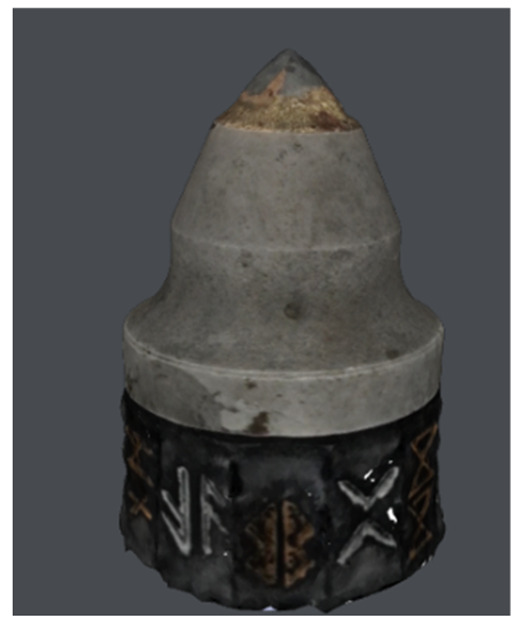	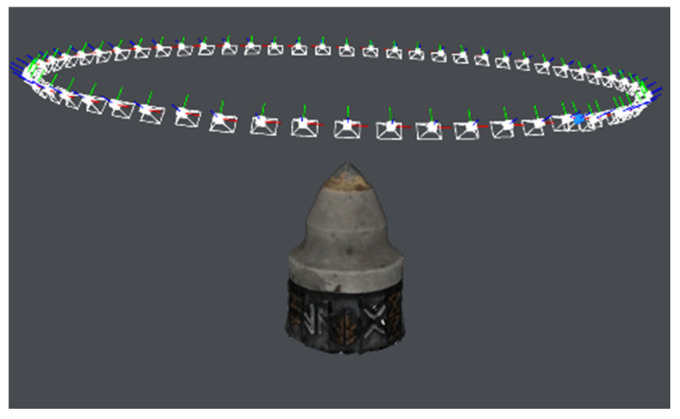
7	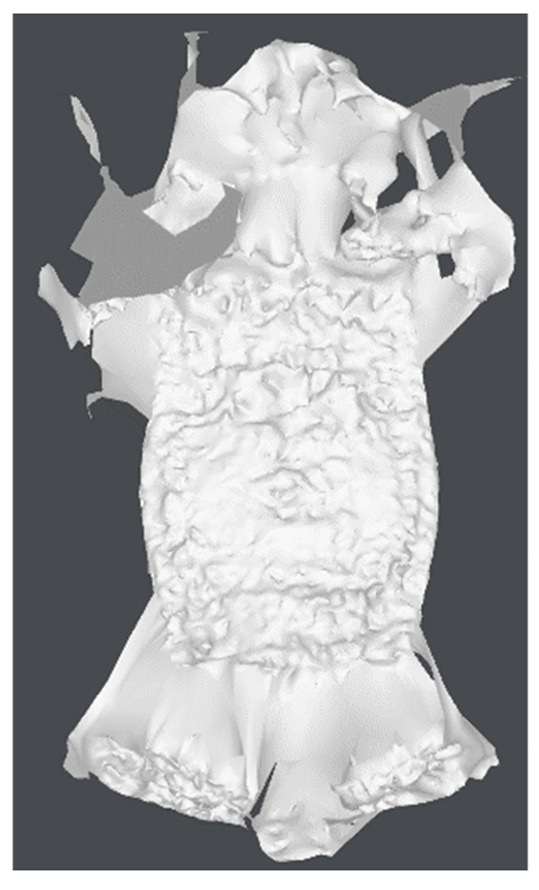	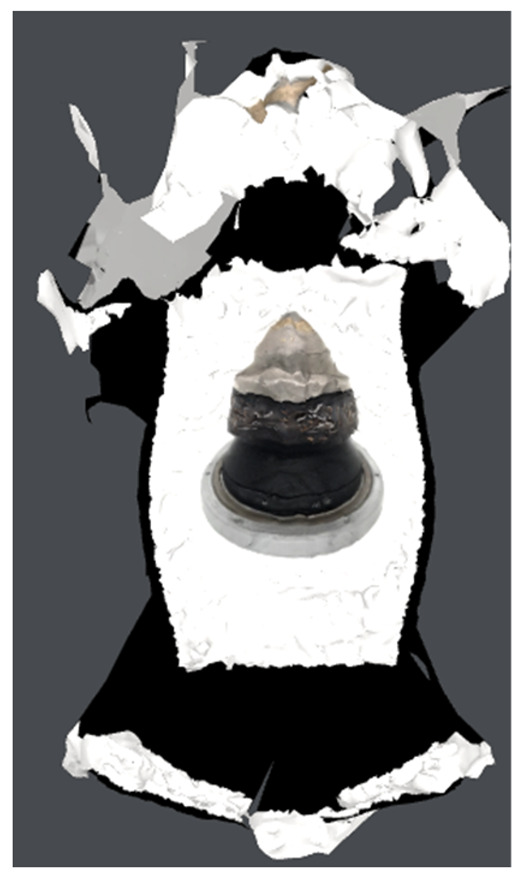	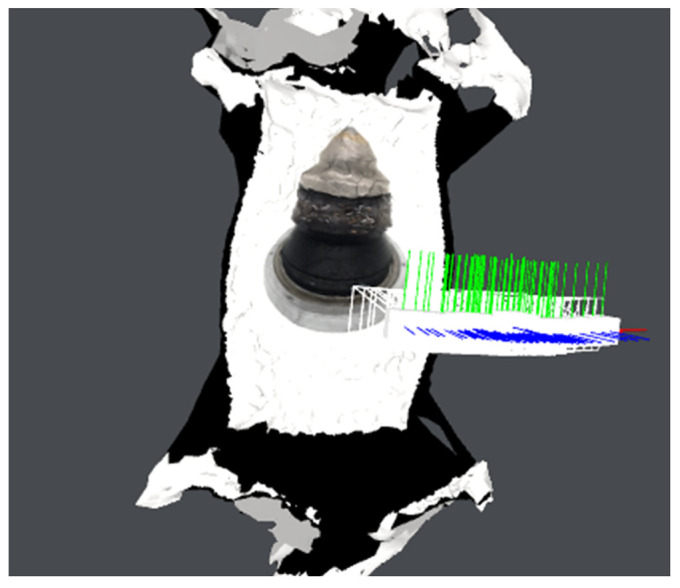

**Table 6 materials-15-05783-t006:** Table containing the calculated data regarding the examined picks. The S parameter is a Boolean value assigned according to set threshold—in this case, the authors set the critical value at 3%. The threshold value may vary in the case of different tool geometry. The maximum plastic deformation coefficient calculates the area of surplus material, exceeding the contour of the master model cross-section.

Pick	Mean [cm^2^]	Std Dev. [cm]	Max. Plastic Def. Area [cm^2^]	Max. Area Diff. [cm^2^]	Area Diff. as a Part of Mean Area [%]	S (Symmetrical Parameter)
CP0	19.869	0.03048	0	0.1080	0.5	1
CP1	18.911	0.038211	0	0.1480	0.7	1
CP2	18.496	0.11077	0	0.3470	1.9	1
CP3	18.416	0.22452	0	0.6360	3.5	0
CP4	17.791	0.48366	0.4910	1.3490	7.5	0

**Table 7 materials-15-05783-t007:** Limits of the methods of classification of the picks’ wear and its features.

Type of Scanning Method	Implementation in a Difficult Environment	Automation Possibilities	Enough Output Data for Regeneration
C2 parameter	+	+	−
Parametric factors	−	−	+
Fuzzy neural network	+	+/−	−
LIDAR measurements	+	+/−	−
Photogrammetric model	+	+	+

## Data Availability

The data presented in this study are available upon request from the corresponding author.

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
