# Peer review of "The Volumetric Wear Assessment of a Mining Conical Pick Using the Photogrammetric Approach"

_materials, 2022, doi:10.3390/ma15165783_

Round 1

Reviewer 1 Report

This paper presents a methodology that is used to create a 3D photogrammetric model. Overall, this paper is well organized, and some promising results are presented. Here are some suggestions for further improving the quality of this paper.

1) It is suggested to have more discussions on the existing research works.

2) Wear is inevitable during the service life of the tool. Therefore, it is suggested to introduce the impact of tool wear and the specific wear mechanisms. The wear mechanisms can refer to 10.1016/j.ymssp.2022.109605. There are some introductions on the wear mechanisms.

3) It is suggested to introduce the whole organization of this paper at the end of the Introduction.

4) It is suggested to have a Conclusion section.

Author Response

Dear Reviewer,

Thank you very much for taking the time to read our manuscript thoroughly and make recommendations for its correction and improvement. We have read the comments carefully and have responded to all your comments.

Remark 1

It is suggested to have more discussions on the existing research works.

Response: Thank you for the remark. The discussion section has been supplemented.

Remark 2

Wear is inevitable during the service life of the tool. Therefore, it is suggested to introduce the impact of tool wear and the specific wear mechanisms. The wear mechanisms can refer to 10.1016/j.ymssp.2022.109605. There are some introductions on the wear mechanisms.

Response: Thank you for the paper, it indeed contains a comprehensive knowledge about wear mechanisms – we added it to the introduction section.

Remark 3

It is suggested to introduce the whole organization of this paper at the end of the Introduction.

Response: We added a paragraph in the “introduction” section.

Remark 4

It is suggested to have a Conclusion section.

Response: Thank you for the remark, the Conclusion section is added.

Reviewer 2 Report

This paper proposes a volumetric wear assessment method of tool by using photogrammetric approach, including both modeling and experiments. The work is interesting and very meaningful in industry, which can be published after revision. Some suggestions are given as follows:

1. How about the efficiency of the proposed method? Which is very important in practice?

2. How to deal some disturbances for imaging from environment? In other words, how about the robustness of the method?

3. The axes of all figures in this paper don’t have units, which is non-standard.

4. The conclusion part is necessary, and some quantitative results should be highlighted in the conclusion part.

5. Besides mining conical pick, many machine tools also use other tools, and the work can be also used for other machine tools to predict wear assessment, which is meaningful for machine tools. This potential advantage can be mentioned in introduction, and the related papers about machine tool and tool wear can be cited:

a) Developing a ball screw drive system of high-speed machine tool considering dynamics. IEEE Transactions on Industrial Electronics, 2022, 69(5): 4966-4976.

b) Online tool wear monitoring via hidden semi-Markov model with dependent durations. IEEE Transactions on Industrial informatics, 2018, 14(1): 69-78.

Author Response

Dear Reviewer,

Thank you very much for taking the time to read our manuscript thoroughly and make recommendations for its correction and improvement. We have read the comments carefully and have responded to all your comments.

Remark 1

How about the efficiency of the proposed method? Which is very important in practice?

Response: The efficiency at the current level of scanning setup development is lower than the C2 mass wear parameter, however it can be improved by adding additional cameras in the “lightbox”. Moreover, this method can diminish the error of wear assessment by reducing the scanned volume of plastic deformation – the traditional mass loss measurements are unable to exclude those deformed pieces of material.

Remark 2

How to deal some disturbances for imaging from environment? In other words, how about the robustness of the method?

Response: It is necessary to enclose the scanned object in box which can isolate the scanned part from the environmental light. The Authors tried to scan the picks outside the “lightbox”, yet the results were poor, comparing to the outcome from “lightbox”-enclosed scanning. Additionally, the lightbox had its own source of light, which could be covered in polarizing film. This improved the scans, especially in case of the reflective surfaces.

Remark 3

The axes of all figures in this paper don’t have units, which is non-standard.

Response: Thank you for the remark, however the figures which represent values with units (such as cm3, for instance) have them included. Fig. 5 and 6 deals with unitless indicators, whereas Fig 7 and Fig 8 use unitless coordinates – those coordinates are later scaled according to the distance from camera to the focal point on the object. We added unitless indicators to the axes.

Remark 4

The conclusion part is necessary, and some quantitative results should be highlighted in the conclusion part.

Response: Thank you for the remark, we have added the conclusion part. The quantitative results are included in Result section.

Remark 5

Besides mining conical pick, many machine tools also use other tools, and the work can be also used for other machine tools to predict wear assessment, which is meaningful for machine tools. This potential advantage can be mentioned in introduction, and the related papers about machine tool and tool wear can be cited:

  1. a) Developing a ball screw drive system of high-speed machine tool considering dynamics. IEEE Transactions on Industrial Electronics, 2022, 69(5): 4966-4976.
  2. b) Online tool wear monitoring via hidden semi-Markov model with dependent durations. IEEE Transactions on Industrial informatics, 2018, 14(1): 69-78.

Response: Thank you very much, we have included those valuable references in the introduction section.

Round 2

Reviewer 2 Report

The paper can be accepted.